# Metabolic Syndrome and β-Oxidation of Long-Chain Fatty Acids in the Brain, Heart, and Kidney Mitochondria

**DOI:** 10.3390/ijms23074047

**Published:** 2022-04-06

**Authors:** Alexander Panov, Vladimir I. Mayorov, Sergey Dikalov

**Affiliations:** 1Department of Biomedical Sciences, Mercer University School of Medicine, Macon, GA 31201, USA; mayorov_vi@mercer.edu; 2Division of Clinical Pharmacology, Vanderbilt University Medical Center, Nashville, TN 37232, USA; Sergey.dikalov@vanderbilt.edu

**Keywords:** metabolic syndrome, kidney, heart, brain mitochondria, β-oxidation, long-chain fatty acids, oxidative stress, human postembryonic ontogenesis

## Abstract

We present evidence that metabolic syndrome (MetS) represents the postreproductive stage of the human postembryonic ontogenesis. Accordingly, the genes governing this stage experience relatively weak evolutionary selection pressure, thus representing the metabolic phenotype of distant ancestors with β-oxidation of long-chain fatty acids (FAs) as the primary energy source. Mitochondria oxidize at high-rate FAs only when succinate, glutamate, or pyruvate are present. The heart and brain mitochondria work at a wide range of functional loads and possess an intrinsic inhibition of complex II to prevent oxidative stress at periods of low functional activity. Kidney mitochondria constantly work at a high rate and lack inhibition of complex II. We suggest that in people with MetS, oxidative stress is the central mechanism of the heart and brain pathologies. Oxidative stress is a secondary pathogenetic mechanism in the kidney, while the primary mechanisms are kidney hypoxia caused by persistent hyperglycemia and hypertension. Current evidence suggests that most of the nongenetic pathologies associated with MetS originate from the inconsistencies between the metabolic phenotype acquired after the transition to the postreproductive stage and excessive consumption of food rich in carbohydrates and a sedentary lifestyle.

## 1. Introduction

Physiologists have long ago established that fatty acids are the primary substrates for deriving energy in the human body. In some organs, namely the heart and kidney, β-oxidation of fatty acids provides more than 90% of ATP [1,2]. On the other hand, there is a strong belief that glucose is the only source of ATP in the central nervous system [3,4]. Although abnormalities of both lipids and glucose metabolisms have direct connections with the metabolic syndrome and the associated pathologies, there is much confusion about how these energy substrates metabolize and interact at the mitochondrial level to produce ATP [3,4,5,6].

Physicians have noticed that many patients after 45–50 acquire some typical features of external appearances and biochemical indices of abnormal metabolism. Some patients acquire abnormalities in carbohydrate metabolism with insulin resistance, whereas others have symptoms of cardiovascular pathologies, but all patients have excessive body weight. In 1981, the publication on the close relations between insulin resistance and abnormalities in lipid metabolism [5] stimulated research in the related fields. By the end of the 1980s, the term “metabolic syndrome” had been accepted as a separate nosological unit. The metabolic syndrome (MetS) diagnosis requires simultaneous presence of at least three out of several simultaneously existing medical conditions: central (visceral) obesity, high blood sugar with insulin resistance, high serum triglycerides, low serum high-density lipoprotein (HDL), and high blood pressure [6,7,8]. These interrelated conditions share underlying mediators, mechanisms, and metabolic pathways [6,7]. Physicians quickly appreciated the practical significance of defining MetS all over the world because it allowed the identification of high-risk patients with atherosclerosis, cardiovascular diseases, hypertension, and type 2 diabetes. However, an analysis of the tremendous number of publications regarding MetS revealed that more than 30 years since defining MetS, there is no deep understanding of how and why MetS develop. Much of the literature can be roughly divided into supporters and opponents of insulin resistance as the primary symptom for diagnosing MetS [8,9]. The controversies are heated, by publications showing that ethnic and racial factors significantly affect the criteria for metabolic syndrome.

## 2. Origin and Features of Metabolic Syndrome

### 2.1. Metabolic Syndrome Represents the Beginning of the Last Stage of Human Postembryonic Ontogenesis

Genetic predisposition [10,11], sedentary lifestyle and low physical activity [12], excessive alcohol use [13], and even environmental factors [14,15] have all been mentioned as possible causes of MetS,. Undoubtedly, aging is the primary factor in the development of MetS [16]. Women before menopause with metabolically healthy overweight/obese phenotype display a better overall metabolic profile and less oxidative stress than that observed in normal-weight individuals with MetS after menopause [17]. In general, obesity per se is not associated with MetS [18,19].

Stančakova and Laakso summarized the results of genetic studies on MetS [20]. They concluded that although several candidate genes regulating primarily lipid metabolism, adiposity, or insulin resistance are associated with multiple MetS-related phenotypes, they provide only limited evidence for shared genetic background explaining the clustering of metabolic traits. However, growing evidence suggests the importance of epigenetic mechanisms [20]. This conclusion, together with the observations that aging is the permanent condition for the development of MetS, support our earlier suggestion [21,22] regarding the aging population of the northern hemisphere, that the external appearances and metabolic features of MetS reflect the genetic properties of our distant ancestors, which resemble those of the contemporary indigenous people of the north.

Aging and MetS have been commonly regarded as the accumulation of various damages caused by oxidative stress and improper lifestyle [12,13,23]. Importantly, aging, first of all, is also the process of growing older or changing over time [24]. After birth, all human beings during postembryonic ontogenesis undergo changes from one stage to another controlled by different hormones and genes. MetS usually develops when an individual enters the postreproductive stage, which usually happens at the age of 50–55 when a sedentary lifestyle and excessive body weight in men may significantly accelerate the development of MetS [12].

Since our predecessors’ life span was concise for hundreds of thousands of years, there was no or only weak natural selection pressure on the genes governing organisms in the postreproductive stage of ontogenesis. For this reason, people, after entering the postreproductive stage, are under the control of genes inherited from their distant ancestors. What was the kind of life of our ancestors? They often starved, were poorly protected from cold and other external conditions, did not consume pasta, sugar, and had no domestic animals and agriculture. They had to work hard to hunt or find food, which could differ in different geographical locations depending on environmental conditions. As a result, the existing mitochondrial DNA haplotypes currently reflect different ways of metabolic adaptation of ancient populations to different conditions and affect the manifestation of the MetS symptoms. From the presented point of view, the manifestation of obesity or insulin resistance as the leading symptoms in MetS may depend on the patient’s nuclear (nDNA) and mitochondrial (mtDNA) background, which created the metabolic phenotype adapted to prolonged famine, primary consumption of fats and proteins but which does not require external glucose. It can explain variations in the prevalence of the MetS-associated diseases observed between populations with different mtDNA haplotypes [21,22,25].

We are far from the first to present the hypothesis mentioned above. In 1962, Neel [26] had put forward the “thrifty gene” hypothesis that some populations may have genes that determine increased fat storage, which in times of famine represented a survival advantage, but in a modern environment results in obesity and type 2 diabetes. To this day, the initial idea of Neel was modified in that it is not a single “thrifty” gene, but many nuclear and mitochondrial genes are forming specific metabolic phenotypes inherited from our predecessors that come into conflict with the contemporary diets and lifestyle [27,28]. In the past decades, insulin resistance and type 2 diabetes have increased at an alarming rate in all Western countries, particularly in countries where indigenous northern populations are adopting a “Western lifestyle”. Carulli et al. (2005) suggested the impact of environmental factors, such as diet, obesity, and low physical activity, on the pathogenesis of diabetes [28]. However, numerous data show that the prevalence of the MetS symptoms and variation of metabolic features depend on environmental conditions and thus differ in various ethnic groups [25,29,30,31,32,33].

### 2.2. Gender Differences in Energy Metabolism Affect the Rate of Appearance of the Metabolic Syndrome

In men, the metabolic transition from the reproductive stage to the postreproductive stage of ontogenesis may begin earlier than the age of 50 and is not as sharp as in women. Initially, some MetS symptoms in men, such as visceral obesity, dyslipidemia, and others, may primarily be associated with excessive food, alcohol, and a sedentary lifestyle, which is true but is not the whole truth. The much sharper development of the MetS symptoms in women gives much more specific information about the essence of the metabolic features of the postreproductive stage of ontogenesis.

The sharp changes in appearance and metabolism are particularly evident in women when they enter menopause, which increases the risk of MetS by 60% [33]. It is crucial that the emergence of MetS in the postmenopause period does not depend on the body mass index and physical activity [34] but may depend, in women, on the dynamics of estrogen decline with age [35]. Interestingly, studies on sex hormone replacement in animals have shown that males receiving testosterone showed MetS deterioration, while females with estrogen replacement showed improvement in their MetS symptoms, such as decreased hypertension [36].

There are observations that women are relatively protected against cardiovascular diseases (CVD) before menopause. The reasons for this sex difference are not entirely understood, but fatty acid metabolism may play a role [37]. Studies of fatty acids metabolism in young men and women have shown that in comparison with men, the partitioning of fatty acids in young women occurs toward ketone body production rather than very-low-density lipoproteins–triacylglycerides (VLDL-TAG), which may contribute to their more advantageous metabolic profile, as compared with young men [38]. However, the postmenopausal status had a pronounced effect on the characteristics of small VLDL particles, which were considerably enriched in triglycerides [39]. People living in the northern countries aged over 50 often develop visceral obesity with a bulky body and other signs resembling their distant ancestors, which are similar to characteristics of the contemporary indigenous populations of the north. Thus, we can conclude that men and women aged 50–55 switch to metabolism similar to our distant ancestors when fats and proteins were the primary energy sources. From this point of view, symptoms of MetS, for example, insulin resistance, reflect this transition.

Some of the MetS-associated pathologies, such as dyslipidemia and type 2 diabetes (T2D), may arise from the inconsistencies between the metabolism adapted to fasting, food rich in fats and proteins and active life, and the “new habits” of excessive consumption of food rich in carbohydrates and sedentary lifestyle. Naturally, in different parts of the world, variations of MetS are expected to be associated with variations in mtDNA haplotypes. That is, probably, why Neel could not find a “thrifty” gene among Central and South American inhabitants. In contrast, Hegele and colleagues found a “thrifty” gene relatively quickly among the indigenous people of Canada, in whom many genes belong to the “thrifty” metabolic phenotype.

## 3. Sex-Specific Differences in the Rates of Aging and Longevity

The effect of sex and recombination increases the efficiency of natural selection, which is a significant factor favoring evolution [37,40]. It has been experimentally shown that sex increases the rate of adaptation to a new harsh environment but has no measurable effect on fitness in a new benign environment with small selection [41].

The data accumulated show that in many species, including humans, females have a slower rate of aging and longer life span than males [42,43,44]. We suggest that this observation has essential general biological goals for females: bearing and raising a new generation despite external difficulties and metabolic restrictions. These goals demand that females have to be efficient but not superefficient from the metabolic (thermodynamic) point of view, resilient to harsh environmental conditions, and have a lower rate of oxidative stress. Numerous studies of various species showed that females generally have slower ROS production than males [41,42].

### 3.1. Sex-Specific Differences in the Rate of Fat Utilization for Production of Energy

Laboratory animals are an indispensable part of biomedical research and are widely used for modeling physiological and pathological situations in humans [45]. It is impossible to study human biomedical problems for ethical and technical restrictions, whereas animal research provides a degree of experimental control and precision not usually feasible in studies using human subjects [46]. At the same time, the animals used in most experiments were males because researchers usually avoided using females due to their reproductive cycles and seasonal hormone fluctuations that could affect the results of their studies [47]. For these reasons, research on sex differences began relatively recently, but the related literature is enormous today. Human studies on metabolic differences between men and women were stimulated mainly due to the progress of sports medicine. Here, we will discuss only those works that directly relate to our subject: sex metabolic differences underlie the fact that females live longer than males in many species, including humans [46].

The sex differences in the body structure and metabolism depend on the stage of a person’s ontogeny. Vijay et al. (2015) studied sexual differences in the expression of mitochondria-related genes in rat hearts at different ages that correspond to different stages of the rat’s reproductive capacities [48]. The authors divided animals into groups of young (8-week), adult (21-week), and old (78-week) male and female Fischer 344 rats and analyzed the expression of 670 unique genes related to various mitochondrial functions. A significant (*p* < 0.05) sexual dimorphism in expression was observed in young animals for 46, adult for 114, and old rats for 41 genes, respectively [48]. Importantly, sexual dimorphism was not noted in genes encoding oxidative phosphorylation in young and adult hearts. Adult males showed higher expression of genes associated with the pyruvate dehydrogenase complex than females. In old rats, most genes involved in oxidative phosphorylation had higher expression in females. The results clearly show that sexual dimorphism largely depends on the stage of ontogeny. Other studies demonstrated better preservation of myocardial mass and greater cardiac contractility in women than men during aging [49]. Better heart health in aged women might result from genetically predetermined factors, less oxidative damage, and slower aging than men.

It is a well-known observation that women generally have higher body fat than men. Women store fat predominantly in the gluteal-femoral region, whereas men have more body fat in the abdominal (visceral) region [50,51]. Notably, multiple endocrine perturbations accompany visceral fat accumulation, including elevated cortisol and androgens in women and low growth hormone and testosterone secretion in men. The hormones’ effects are more evident in visceral than subcutaneous adipose tissues because omental fat has higher cellularity, innervation, and blood flow. Furthermore, the density of cortisol and androgen receptors is higher in visceral fat than in other adipose tissue regions [52]. In addition, there are epidemiological and metabolic associations between visceral fat accumulation and disease [52], which is vital because visceral obesity is a common symptom for men and women with metabolic syndrome.

The experiments in vivo indicate that sex-specific hormones control sex variations in fat metabolism. We can presume that in the postreproductive stage of ontogenesis, the sexual dimorphism in fat metabolism should be weaker or absent. Results of studies in vitro also indicate that this difference is diminished at menopause and may be restored by estrogen therapy [52]. These facts suggest that estrogens’ functional effects in women are similar to those of testosterone in men. As we will show in the next section, the effects of both hormones are targeted on substrate oxidations by mitochondria and, thus, on the rates of ROS production. However, the mechanism of estrogen on fat metabolism might be indirect because human adipose tissue does not possess specific estrogen and progesterone receptors [52].

No gender differences in the basal level of the net muscle protein balance have been found [53]. In general, testosterone increases muscle protein synthesis and increases muscle mass. At a young age, boys and girls have similar amounts of testosterone. Testosterone levels increase dramatically in males at puberty, as does muscle mass.

### 3.2. Sex Differences in Substrate Utilization during Physical Activities

The current data on sexual dimorphism in utilizing fatty acids during physical activities support the effects of sex hormones on fat metabolism. They demonstrate that the proportion of energy derived from fat during exercise is higher in women than in men [50,54]. Carter et al. (2001) investigated the effect of endurance training on a whole-body substrate, glucose, and glycerol utilization during 90 min of exercise in males and females [54]. First, females show a lower respiratory exchange ratio (RER) than males during submaximal physical loads, indicating a proportionately lower carbohydrate and higher fat oxidation [54,55]. Compared with females, males significantly increase their need for amino acids to fuel energy needs during intensive physical activity. Under the same conditions, females responded by increased fat mobilization, requiring fewer alternative fuels, such as carbohydrates and amino acids [54,55,56]. The overall conclusion of these experiments was that females oxidize a more significant proportion of fat and fewer carbohydrates and amino acids than males. Thus, physiologists indirectly support our finding that β-oxidation of fatty acids requires the simultaneous presence of other mitochondrial metabolites derived from carbohydrates or proteins [57]. Because women have a lesser consumption of substrates supporting the β-oxidation of fatty acids, the rate of fatty acid oxidation should also be lower and thus less effective. This suggestion might explain why, in general, women demonstrate lower physical performance levels during endurance sports and produce less ROS [58].

Mitochondria oxidize fatty acids only with supporting substrates [57,59]. The data presented above suggest that females require less or different supporting substrates for the effective oxidation of fatty acids. Unfortunately, until now, we have been the only ones who have studied the oxidation of fatty acids in the presence of various supporting substrates, and we used male rats only in our experiments [57,60]. Therefore, the experiments on gender dependence of mitochondrial oxidative phosphorylation are awaiting fulfillment.

Figure 1 illustrates how the rates of ROS production by the isolated rat heart mitochondria depend on the supporting substrates. Thus, the in vitro experiments with the isolated mitochondria from different organs require further investigation to elucidate the molecular mechanisms of sex diversity at the mitochondrial level. Since males showed higher expression of genes associated with the pyruvate dehydrogenase complex as compared to females [49] and consumed more carbohydrates and amino acids during endurance training [54,55,56], we suggest that women’s higher longevity and a slower rate of aging are associated with the less efficient oxidation of fatty acids and thus slower rates of oxidative stress. However, this conclusion regards only women at the reproductive stage, when women are “protected” [61]. Ventura-Clapier et al. (2020) stated that “apart from comparisons between males and females, there is a crucial need to study female physiology and woman pathology. In particular, the biological step that constitutes menopause in women appears to be the border between “female protection” and “female susceptibility” to cardiovascular diseases, which needs to be deciphered further” [61].

### 3.3. Gender Differences in the Transition from Reproductive to the Postreproductive Stage

Olivetty et al. [49] studied changes in mononucleated and binucleated myocytes with age in enzymatically dissociated cells. The age interval examined varied from 17 to 95 years. The authors found that in the course of aging, women’s hearts preserved the ventricular myocardial mass, the aggregate number of mononucleated and binucleated myocytes, average cell diameter, and volume. In contrast, in the men’s hearts, the authors observed loss of myocardium at the rate of nearly one g/year, and this phenomenon accounted for the loss of approximately 64 million cells. These detrimental events involved the whole male heart. In the remaining cells, myocyte cell volume increased at 52 microns/year in the left and 56 microns/year in the right ventricle. Moreover, these changes in men’s hearts were linear from the age of 17 to 95, whereas, in women, the structural properties of the heart remained unchanged [49]. Thus, it seems that women enter the postreproductive stage with relatively “young” hearts, whereas in men, the aged heart loses many cells, and the remaining cells increase their volume, which can be a disadvantage for the heart’s energy metabolism.

Figure 2 summarizes the available information on approximate differences in the rates of ROS production, which determines the rate of aging during ontogeny. The graphic presented in Figure 2 visualizes the long-known fact that women age slower than men until menopause, when women’s aging rate sharply increases. What are the metabolic features responsible for this dramatic gender difference in the rate of aging?

The mitochondrial free radical theory of aging, initially proposed by Harman [23], is the most relevant to the problems of gender variations in the development of metabolic syndrome, aging, and associated pathologies [60].

## 4. β-Oxidation of Fatty Acids Is the Primary Source of Energy in Humans

The carbohydrate storages consist of a limited amount of blood glucose and glycogen in the human body and have 2.44 times lower caloric value than acyl fatty acids [60]. In the human body, several functions are dependent on glycolysis. Blood erythrocytes have no mitochondria and thus rely only on glycolysis to maintain their energy demands. In an average human, with 5 L of blood containing 5 g of glucose, erythrocytes consume a minimum of 240 g of glucose a day (A. Panov, unpublished data). Much glucose is used, particularly in growing children, for synthesizing nucleic acids, purine, and pyrimidine nucleotides. In the central nerve tissues, glycolysis provides energy for propagating the electrical signals along the axons, whereas the synaptic mitochondria can perfectly well oxidize short- and long-chain fatty acids [59]. Thus, glucose is too precious to be used as the primary energy source. Because the storage is limited, glucose is constantly produced by the liver and kidneys [62]. In addition, in the hard-working organs, such as the heart, kidney, brain, and skeletal muscles, pyruvate oxidation via the mitochondrial NADH-oxidase (complex I) is the rate-limiting step. It thus cannot maintain fast ATP-consuming functions in the brain, heart, and kidneys [60,63]. Only oxidation of fatty acids can provide fast provision of ATP and support anaplerotic metabolic pathways for prolonged periods.

The energetic efficiency of β-oxidation of long-chain fatty acids in the presence of supporting substrates is the only combination of substrates capable of supporting the highest rates of oxidative phosphorylation in the heart during maximal physical loads [57,59]. During β-oxidation, there is a simultaneous reduction in the NADH/NAD^+^ system in mitochondria and the membrane pool of ubiquinol/ubiquinone. Accordingly, the electrons can enter the respiratory chain not only via the large supercomplexes of the respirasome, which contain complex I, but they can enter much faster via the smaller supercomplexes containing one dimer of complex III and two dimers of complex IV (Figure 3). In addition, electrons from CoQH_2_ enter the respiratory chain by reversing the succinate dehydrogenase reaction (complex II) [64,65].

During β-oxidation of fatty acids, the electrons enter the respiratory chain simultaneously through complexes II and III, and, to a lesser extent, through complex I, which allows mitochondria in the heart, kidney, and brain to maintain high rates of respiration and ATP provision even at the maximal functional loads in these organs. The heart, brain, and skeletal muscles have a wide range of functional activities. Therefore, when the ATP consumption diminishes, the excess of energy in mitochondria may redirect electrons to produce the superoxide radicals [57,60,64], and thus, HO_2_^•^, the central oxygen radicals responsible for oxidative stress, and thus, aging [68].

## 5. Properties of β-Oxidation of Long-Chain Fatty Acids and Generation of Superoxide Radicals by the Kidney, Brain, and Heart Mitochondria

The enzymes of the tricarboxylic acid (TCA) cycle and fatty acids β-oxidation, which provide hydrogen for the respiratory chain complexes, are also structurally associated with the respiratory complexes of the inner mitochondrial membrane [69,70,71]. For example, the pyruvate dehydrogenase and α-ketoglutarate dehydrogenase complexes associated with the inner membrane exceed those in the matrix 200 times [72]. The close interactions of the metabolic enzymes with the respiratory complexes are necessary for functional flexibility providing ATP in a wide range of functional activities typical for the heart, brain, and skeletal muscles [73]. It is known that respiring mitochondria generate superoxide radicals only during resting respiration (State 4) or when the capacity to generate membrane potential exceeds the rate of ATP consumption [64]. Although mitochondria can generate ROS at many different sites, the most significant production occurs on Complex I during the reverse electron transport (RET) mediated by succinate [63,64]. RET is an energy-dependent process and is accompanied by a significant increase in the rate of State 4 respiration, which is particularly large during the β-oxidation of fatty acids. We can suggest that when CoQH_2_ interacts with the small respiratory complex (see Figure 3), there is no production of ROS because the electrons are instantly transported to Complex IV in a highly exergonic reaction. However, the electrons transported via the reversal of the Complex II (SDH) reaction will reduce and thus produce ROS on the flavin adenine nucleotide moiety of complex II [64,65], and in the energized mitochondria, they will stimulate RET and reduce the numerous sites producing ROS on Complex I [64,65,74].

### 5.1. Intrinsic Inhibition of Succinate Dehydrogenase (Complex II) as a Protection against Excessive Production of ROS

The isolated brain and heart mitochondria possess a potent intrinsic inhibition of the succinate dehydrogenase, thus preventing ROS generation associated with RET in resting mitochondria [74]. Figure 4 demonstrates the oxygen consumption rates by the isolated kidney, brain, and heart mitochondria oxidizing succinate in the absence of rotenone. One can see that the isolated mitochondria from the brain and heart oxidize succinate poorly, particularly in the presence of ADP. Adding glutamate releases the SDH inhibition by removing oxaloacetate from the active center in the transaminase reaction (Figure 4B,C) [74].

Kidney mitochondria (Figure 4A) have no intrinsic inhibition of SDH (Complex II), and thus, both respiratory rates in the metabolic States 4 (resting respiration) and 3 (active phosphorylation of ADP) are high [75]. We suggest that, unlike the skeletal muscle, heart, and brain mitochondria, which have an extensive range of workloads, the kidney mitochondria work at all times more or less evenly, do not have resting states, and thus have no intrinsic SDH inhibition.

In conclusion of this section, it should be noted that the degree of the intrinsic inhibition of succinate oxidation is subject to significant variations between organs and species in particular [74].

### 5.2. β-Oxidation of Long-Chain Fatty Acid by the Isolated Kidney, Brain, and Heart Mitochondria

Among numerous metabolites, which have been used as respiratory substrates by researchers for the isolated mitochondria, β-oxidation of the long-chain fatty acids has been the least studied. The primary reason is that the respiratory activities with L-acyl-carnitines were poor with the isolated mitochondria from the heart, brain, and other organs. Because with acyl-carnitines as a substrate there is a significant increase in the rate of the State 4 respiration, researchers regarded long-chain fatty acids as uncouplers. As for the brain mitochondria, some researchers concluded that brain mitochondria do not oxidize long-chain fatty acids and recommended short-chained fatty acids as substrates [3,4,76]. However, when we started to utilize a mixture of L-acyl-carnitine with other mitochondrial substrates, namely glutamate, succinate, and pyruvate, we observed very high rates of respiratory activities in all isolated mitochondria, including the brain synaptic mitochondria [57,59,74]

Figure 5 demonstrates how various supporting substrates affect the oxidation of L-palmitoyl-carnitine + malate by the kidney, brain, and heart mitochondria during resting respiration (State 4).

As shown in Figure 5 and earlier publications [57,59], during resting respiration oxidation of palmitoyl-carnitine in the presence of supporting substrates, there is a several-fold increase in the oxygen consumption rates in the kidney, brain, and heart mitochondria. The significant increases in oxygen consumption during the State 4 respiration are caused not by the uncoupling, as was suggested earlier, but by the much more effective energization of the mitochondria and activation of the reverse electron transport, which occurs only in the highly energized mitochondria [57,59]. The data presented in Figure 6 support this conclusion: the rates of oxidative phosphorylation increase several-fold when the mitochondria oxidize palmitoyl-carnitine in the presence of supporting substrates. Classical uncouplers cause dissipation of the mitochondrial transmembrane potential as heat, thus diminishing the rate of ATP production.

As a supporting substrate, Figure 5 and Figure 6 demonstrate that succinate provides the highest rates of oxidative phosphorylation with the kidney, brain, and heart mitochondria oxidizing palmitoyl-carnitine. There is evidence that succinate stimulates the reverse transport of electrons from the membrane pool of CoQH_2_ into the respiratory chain [64,65]. Panov et al. [75] presumed that the stimulatory effect of succinate on β-oxidation of palmitoyl-carnitine can be allosteric, since the effect was pronounced at low (0.5 mM) succinate concentration. We have also suggested that the supporting effects of glutamate and pyruvate may be associated with converting these substrates first to α-ketoglutarate and then to succinate via transamination reactions [75].

### 5.3. Generation of Superoxide Radicals by the Kidney, Brain, and Heart Mitochondria

In experiments with the isolated mitochondria, State 4 respiration, that is, in the absence of active oxidative phosphorylation, mitochondria acquire the highest level of energization and the most significant production of ROS, mostly superoxide radicals accompanied by the small amount of hydrogen peroxide [64]. In vivo, mitochondria always phosphorylate ADP; thus, it is correct to say that in vivo resting respiration occurs when the rate of mitochondrial energization is higher than the rate of de-energization due to ATP consumption, and therefore, mitochondria maintain the generation of ROS [64]. In experiments in vitro, the State 4 oxygen consumption rate depends on the intrinsic inner mitochondrial membrane proton and electron leaks [77,78], whereas in vivo, the rates also depend on ATP turnover.

Based on these data, we can suggest that, upon transition to the postreproductive stage of ontogenesis, men’s and women’s metabolism utilizes long-chain fatty acids as the primary energy substrate with amino acids as the supporting substrates. Since the diet of our distant ancestors was based on proteins and fats, insulin resistance must be considered not a pathology but a standard feature of the postreproductive-stage metabolism.

### 5.4. Oxidative Stress, Promoted by Metabolic Syndrome, Is the Primary Damaging Factor for the Heart and Brain but Not for the Kidney

With the development of metabolic syndrome, long-chain fatty acids prevail even more as substrates for energy production. With the discovery of the roles of supporting substrates in the β-oxidation of fatty acids, we can suggest that in women, because of the postmenopause change in genetic background, the supporting fatty acids oxidation substrates become similar to those in men. The shift may result in increased ROS production and an accelerated aging rate. The heart and brain are particularly vulnerable to metabolic syndrome-associated pathologies. It is essential that the heart and brain have a wide range of functional loads and thus have an increased risk of oxidative stress damage at rest. That is why many symptoms of metabolic syndrome, including type 2 diabetes, strongly depend on lifestyle [6,28,79,80,81].

The intrinsic inhibition of SDH (complex II) represents an evolutionary mechanism of diminishing ROS production in those organs, which have a wide range of functional activities and utilize long-chain fatty acids as a predominant substrate for energy production. Kidneys, however, do not possess this mechanism, and this raises the possibility that oxidative stress is not the primary pathogenic mechanism for the kidneys in people with metabolic syndrome.

## 6. The Mechanisms of Kidney Failure under Conditions of the Metabolic Syndrome

The kidneys’ central structural and functional units are nephrons in the kidney cortex [82]. Nephrons consist of glomerulus, where filtration occurs, and tubules, where glucose and cations undergo reabsorption. The kidneys filtrate up to 180–200 L of liquid containing ions and metabolites daily in humans. The process of filtration uses transcapillary difference in hydrostatic and oncotic pressures as a driving force [83]. The central feature of reabsorption in the tubules is the symport of glucose and sodium by the sodium-glucose cotransporter (SGLT). The basolateral activity of Na/K-ATPase drives the symport of glucose and Na^+^, thus facilitating glucose uptake against the intracellular concentration gradient [84]. Basolaterally, glucose exits the cell through facilitative glucose transporter 2 (GLUT2) [85]. Glomerulae and tubules contain 44% mitochondrial volume as a percentage of cytosolic volume [86], which is higher than in the heart, and β-oxidation of fatty acids is the predominant source of ATP necessary for reabsorption [2].

In patients with MetS, the developing renal dysfunction and diabetes-induced renal complications are significant causes of morbidity and mortality. Much of the evidence indicates that oxidative stress plays an essential role in the pathogenesis and evolution of chronic kidney disease, including diabetic nephropathy, IgA nephropathy, polycystic kidney disease, and cardiorenal syndrome [87]. However, growing evidence shows that the primary mechanism that switches the chain of pathogenic mechanisms, including hypertonia and oxidative stress, is hyperglycemia, a common symptom of MetS and the main symptom of T2D, resulting in altered renal oxygen metabolism and a decreased renal oxygen tension [88,89,90,91]. Some researchers recognize renal tissue hypoxia as a unifying pathway to chronic kidney disease [89].

The Diabetes Control and Complication Trial Research Group [92] stressed that an essential predictor of diabetes-induced complications is the degree of hyperglycemia. Because glucose reabsorption is tightly coupled with the symport of sodium [84], we can imagine the following situations:Normally, both glucose, and Na^+^, as symport members, are reciprocal at physiological levels.The concentration of Na^+^ in the glomerular filtrate exceeds glucose. Therefore, for the complete Na^+^ reabsorption, kidneys produce glucose via gluconeogenesis. Since 95.5% of the symport of glucose and Na^2+^ occurs in the proximal S1 and S3 sections of the tubule [89], glucose synthesis must occur in the podocytes.The concentration of glucose in the glomerular filtrate dramatically exceeds that of sodium. In the absence of a reciprocal amount of Na^+^, the untransported glucose becomes excreted with urine. Because glucose is a valuable metabolite for the body, we suggest that one of the mechanisms to diminish glycosuria may be converting glucose into lactic acid in the distal parts of tubules. Lactate is also a valuable metabolite reabsorbed back into the blood. We suggest that a high concentration of lactate in the medullar part of the kidneys shows high glycolytic activity not to produce ATP but rather to save valuable substrate for other organs, such as the liver and the central nervous system, and diminish the level of hyperglycemia.Both concentrations of glucose and sodium in the glomerular filtrate are unphysiologically high. Under this condition, the kidneys must work extremely hard, which results in hypoxia and finally causes kidney dysfunction.

The last two situations are typical for T2D, a common disease in patients with metabolic syndrome.

### 6.1. β-Oxidation of Long-Chain Fatty Acids in the Presence of Supporting Substrates Provides the Highest Rates of ATP Production

Typically, about 99.5% of glucose reabsorption occurs in the S1 and S3 segments of the proximal convoluted tubules [89]. The apical membrane of segment 1 contains the high-capacity, low-affinity SGLT-2, whereas the S3 segment contains low-capacity, high-affinity SGLT-1 [2,88]. Under steady-state conditions, the consumption of ATP by the Na, K-ATPase, which drives the glucose-Na^+^ symport [93,94], is matched by the production of ATP by mitochondrial oxidative metabolism [95]. Thus, the renal mitochondria play an integral role in maintaining the energy-requiring processes of the kidney.

There are recent publications that indicate that impaired β-oxidation of fatty acids results in chronic kidney disease [96], lipogenesis [97], and its long-term sequelae lead to fibrosis [98]. In the literature, however, there is no information about the rates of β-oxidation of the long-chain fatty acids by renal mitochondria. Soltoff [99] suggested that the rate of energy production by tubular mitochondria could be compared with the rate of State 3 respiration by the isolated tubular mitochondria, which theoretically should correspond to the maximal rate of ATP production. However, in the experiments in vitro, the rate of oxidative phosphorylation highly depends on the type of substrate or substrate mixtures provided to mitochondria. In the early experiments on oxidative metabolism of the normal glomerulus, the inadequate selection of substrates for oxidation led to the erroneous conclusion that the oxygen uptake of the glomerulus is small [100].

Harris et al. [101] studied the suspensions of proximal rabbit tubules and observed that the average rate of tubular respiration in the presence of NADH-linked substrates (glutamate + malate) and short-chain fats was about 60% of the State 3 rates of isolated tubular mitochondria. In the presence of increased intracellular sodium concentration, O_2_ (QO_2_) consumption increased to the entire State 3 rates. Under this condition, the Na,K-ATPase activity was limited by the rate of ATP production, and the ATP content was reduced [100]. The exposure of the proximal tubule to short-chain fatty acids enhanced reducing equivalents to the mitochondria when the sodium pump was stimulated. In these experiments, the mean State 3 respiratory rate obtained in tubule preparations at 37 °C was 37.3 ± 1.2 nmol O_2_/min/mg of protein. Simultaneous oxidation by tubule mitochondria of glutamate + malate + valerate (C5 fatty acid) or butyrate (C4 fatty acid) increased the rate of respiration by 20% (45 nmol of O_2_/min/mg of protein). However, valerate had no stimulatory effect on the State 3 oxidation of glutamate + malate by the isolated kidney mitochondria [100].

In our experiments with the isolated kidney mitochondria, the State 3 oxidation rate of glutamate + malate was 93 ± 13 nmol O_2_/min/mg protein (not shown). Figure 6 shows that the oxidation by kidney mitochondria of palmitoyl-carnitine plus glutamate practically did not affect the State 3 respiration rate (100 ± 11 nmol O_2_/min/mg protein). Although glutamate + malate is an effective substrate combination for kidney mitochondria in experiments in vitro, in vivo, mitochondria do not utilize glutamate as the sole or primary substrate for ATP production. Long-chain fatty acids are the primary mitochondrial substrates for the kidney mitochondria [1,2,75]. Figure 6 shows that long-chain fatty acids, in the presence of succinate as a supporting substrate, provide the fastest ATP production rate (171 ± 8 nmol O_2_/min/mg protein) and support the fastest rates of glucose-Na^+^ symport dictated by the Na, K-ATPases’ activity in segment 1. If we consider that the local temperature inside respiring mitochondria is around 50 °C [102], the oxygen consumption rates must be significantly higher than ever reported in the literature.

### 6.2. Mechanism of Kidney Hypoxia Development at Persisting Hyperglycemia

In the heart, skeletal muscles, and brain functioning in a wide range of workloads, an increase in ATP consumption results in increased blood flow through the organ, delivering more oxygen. Energy utilization by the kidney is primarily devoted to NaCl and NaHCO3 reabsorption [89,103]. Therefore, increased O_2_ delivery in the kidney via increased blood flow results in augmented tubular electrolyte load [88,89]. The most active oxygen consumption occurs at the first segments of the proximal convoluted tubules S1 and S3 [104]. Thus, sustained hyperglycemia, particularly if accompanied by high sodium content in the blood, results in intrarenal chronic tissue hypoxia because increased O_2_ consumption becomes unmatched with O_2_ delivery. As we showed earlier, β-oxidation of fatty acids in the presence of supporting substrates may support very high rates of oxidative phosphorylation and thus oxygen consumption.

Unlike other organs, vasoactive substances make the situation even worse in the kidney [88,104,105]. Typically, the oxygen and substrate demands for reabsorption are autoregulated via the renal blood flow in a complex way unique to the kidney. The regulatory mechanisms include the autocrine, paracrine, and renin-angiotensin hormonal systems (RAS) (reviewed in ref [106]). In nephrons, NO, as a part of the paracrine system, and angiotensin II antagonistically control glomerular hemodynamic and reabsorptive activities [104].

Under normal conditions, nitric oxide is a renal vasodilator, which, along with the coordinated influences of other renal vasodilators and vasoconstrictors, regulates kidney oxygen and substrate supply [104]. However, the half-life of ^•^NO radical is about 7 s, and since the formation of one molecule of ^•^NO requires two molecules of oxygen, for the maintenance of a steady-state concentration of 1 µM ^•^NO for 1 min, 1 g of a tissue consumes 120 nanomol O_2_ [107]. Therefore, under the conditions of chronic hypoxia induced by hyperglycemia, the induction of ^•^NO production enhances hypoxia. That is probably why researchers have found that nonselective nitric oxide synthase (NOS) inhibitors can increase QO_2_ in the kidney while reducing the glomerular filtration rate [108].

Chronic kidney hypoxia causes a local activation of the renin-angiotensin system (RAS). This results in constriction of efferent arterioles and hypoperfusion of postglomerular peritubular capillaries. In addition, angiotensin II induces oxidative stress via the activation of NADPH oxidase in the endothelial cells [105]. With the progression of high blood pressure, the hydrodynamic forces acting on podocytes may cause their detachment and damage the glomerular filtration barrier [91,109]. Thus, hypertension contributes to progressive kidney dysfunction, manifested as gradual glomerulosclerosis, interstitial fibrosis, proteinuria, and eventually declining glomerular filtration. Oxidative stress plays an essential role in developing and progressing chronic kidney disease and the development of cardiorenal syndrome [87].

### 6.3. Mechanisms of Oxidative Stress under Conditions of Persistent Hyperglycemia

The deleterious effects of reactive oxygen species in the development of kidney failure were recognized a long time ago [88,89,90,91,109,110,111,112,113,114]. However, only a few papers have discussed the mechanisms of deleterious effects of ROS [104,114,115]. Recently, there were serious changes in understanding the roles of individual radicals in oxidative stress and aging [60,68]. The fastest rates of ROS production occur during the β-oxidation of fatty acids, when electrons can be reversed from CoQH2 via complex II into the respiratory chain [64,65]. The enhanced utilization of fatty acids as a primary energy source in all major organs is one of the significant features of the metabolic syndrome. The above sections suggest that, normally, mitochondria in the constantly working kidney produce a minor reactive oxygen species. However, the situation changes under the conditions of hyperglycemia.

In the kidneys exposed to persistent hyperglycemia, the highest oxygen consumption rates will occur primarily in the S1 and S3 segments of the proximal convoluted tubules. In the mitochondria oxidizing long-chain fatty acids in the presence of supporting substrates, even minor hypoxia will reduce the components of the respiratory chain and thus promote ROS production (see Figure 2). The activation of angiotensin II enhances renal hypoxia via both hemodynamic and nonhemodynamic mechanisms, including ROS generation [89,110]. Acidification of the cellular compartments, which accompanies hypoxia, increases the fraction of the superoxide radical (O_2_^•^) converted into the highly reactive perhydroxyl radical. Perhydroxyl radical (HO_2_^•^) specifically interacts with the polyunsaturated fatty acids, initiating the isoprostane pathway of lipids peroxidation (IPPOL), which produces a large variety of biologically active end products resembling normal prostanes as well as highly toxic molecules of isoketales [68]. For this reason, the consequences of IPPOL are also highly variable and initially appear more like accelerated aging. In the endothelial cells of blood capillaries, an enhanced production of both ^•^NO and O_2_^•^ results in the formation of highly toxic peroxynitrite radical, causing damage to the capillaries and thus further exacerbating hypoxia [116]. Notably, the IPPOL mechanism of oxidative stress and aging is insensitive to treatments with antioxidants [117].

## 7. Conclusions

This review shows that metabolic syndrome represents one of the last stages of a normal human’s postembryonic ontogenesis, namely the postreproductive stage. According to the hypothesis, the genes governing this stage experience relatively weak evolutionary selection pressure, representing the metabolic phenotype of distant ancestors. We consider people whose ancestors lived in the northern hemisphere and whose food consisted of meats and fat with a minimum intake of carbohydrates, similar to the habits of the contemporary indigenous populations of the far north. From this point of view, after the transition to the postreproductive stage, β-oxidation of long-chain fatty acids becomes the primary metabolic pathway to satisfy the body’s energy needs, and insulin resistance is the norm.

Despite the fact that physiologists decades ago recognized long-chain fatty acids as the predominant source of energy for the heart, skeletal muscle, and kidney [1,2,118], at the mitochondrial level, the β-oxidation of long-chain fatty acids remains poorly studied. For this reason, our selection of supporting references was limited. The need for additional studies of long-chain fatty acids as a major energy source was stressed by the recent discovery of the new fundamental roles fatty acids play in human development and functions in endothelial cells [119,120]. Finkel’s laboratory has shown that endothelial fatty acid oxidation (FAO) is a critical in vitro and in vivo regulator of the endothelial-to-mesenchymal transition (EndoMT) required for normal heart valve development. Deregulation of EndoMT leads to a wide range of pathological conditions [119]. The Zoltan Irany group [120] have found that endothelial fatty acid uptake is promoted by fatty acid transport protein 4 residing in the endoplasmic reticulum (ER) and that endothelial ER is intimately juxtaposed with mitochondria. These data indicate that mitochondrial ATP production, but not total ATP levels, drives endothelial fatty acid uptake and transport via acyl-CoA formation in mitochondrial/ER microdomains. We suggest that mitochondrial fatty acid β-oxidation plays an important physiological role beyond the heart and muscle tissue, and the role of dysregulation of mitochondrial fatty acid β-oxidation should be further investigated in the presence of supporting substrates.

Mitochondria oxidize long-chain fatty acids at high rates in the presence of other mitochondrial metabolites, namely succinate, glutamate, and pyruvate. Because heart and brain mitochondria work at a wide range of functional loads, they possess an intrinsic inhibition of succinate dehydrogenase (complex II) to prevent or diminish oxidative stress at periods of low functional activity. Kidney mitochondria work at a high rate most of the time, and they do not have the intrinsic inhibition of SDH. We suggest that in people with MetS, oxidative stress and accelerated aging are the primaries in the pathogeneses of the heart and brain pathologies. We propose that oxidative stress is a secondary pathogenetic mechanism in the kidneys, while the primary mechanisms are kidney hypoxia and hypertension caused by persistent hyperglycemia.

The ontogenetic hypothesis of MetS suggests that most of the nongenetic pathologies associated with MetS originate from the inconsistencies between the metabolic phenotype acquired after the transition to the postreproductive stage and excessive consumption of food rich in carbohydrates and a sedentary lifestyle.

## Figures and Tables

**Figure 1 ijms-23-04047-f001:**
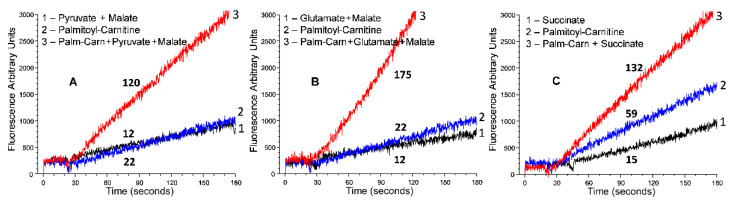
Production of superoxide radicals by rat heart mitochondria oxidizing palmitoyl-carnitine. Designations: **1**. supporting substrate only; **2**. palmitoyl-carnitine only, and **3**. palmitoyl-carnitine + supporting substrate. **Substrates**: (**A**)—pyruvate 2.5 mM + malate 2 mM, (**B**)—glutamate 5 mM + malate 2 mM, and (**C**)—succinate 5 mM. Experimental conditions are described in refs [57,60]. The numbers at the lines are the rates of ROS production in picomol of H_2_O_2_ per minute per mg of mitochondrial protein.

**Figure 2 ijms-23-04047-f002:**
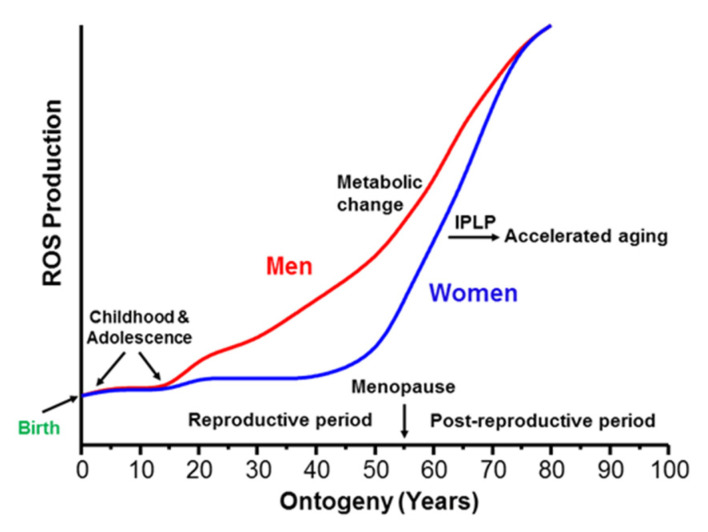
A schematic presentation of approximate differences between men and women in ROS production during ontogeny. The figure was adapted from ref [60].

**Figure 3 ijms-23-04047-f003:**
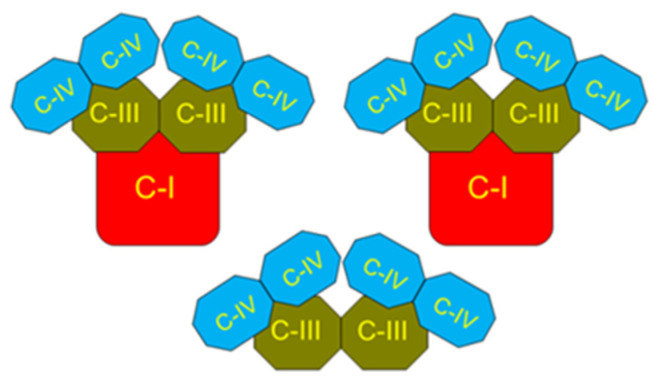
Schematic presentation of the mitochondrial respirasome structure. The figure is based on the data presented in refs [66,67].

**Figure 4 ijms-23-04047-f004:**
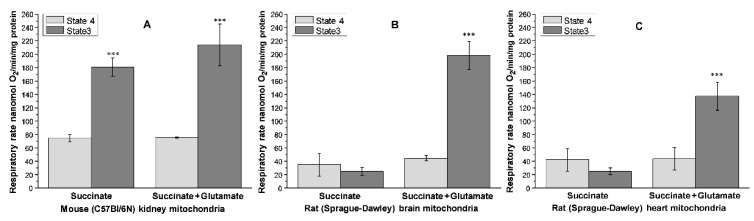
Kidney mitochondria do not possess the intrinsic inhibition of succinate dehydrogenase. Oxygen consumption was measured without ADP (State 4) or with ADP (State 3). (**A**) Mouse kidney mitochondria; (**B**) Rat brain mitochondria; (**C**) Rat heart mitochondria. The data are mean ± standard error. *** *p* < 0.001 vs. State 4 (*n* = 5). Redrawn from ref [75].

**Figure 5 ijms-23-04047-f005:**
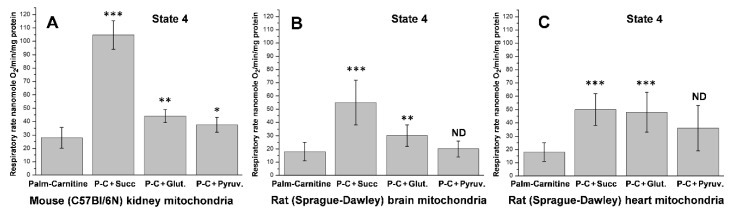
Effects of succinate, glutamate, and pyruvate on the resting oxidation of palmitoyl-carnitine (P-C) by the isolated kidney, brain, and heart mitochondria. Graphs show the oxygen consumption rates without ADP (State 4). (**A**) Mouse kidney mitochondria; (**B**) Rat brain mitochondria; (**C**) Rat heart mitochondria. The data are mean ± standard error. *** *p* < 0.001 Palm-Carnitine (P-C) vs. P-C + Succinate (*n* = 5); ** *p* < 0.01 P-C vs. P-C + Glutamate; * *p* < 0.05 and No-Difference Palm-Carnitine vs. P-C + Pyruvate. Redrawn from ref [75].

**Figure 6 ijms-23-04047-f006:**
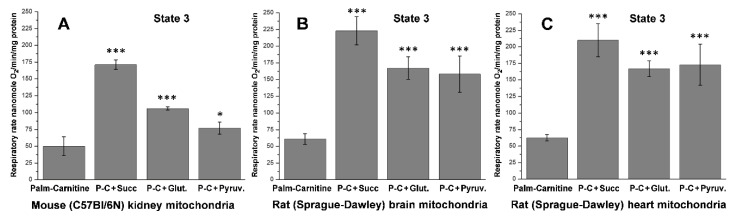
Effects of succinate, glutamate, and pyruvate on the rates of oxidative phosphorylation during oxidation of palmitoyl-carnitine (P-C) by the isolated kidney, brain, and heart mitochondria. Graphs show the oxygen consumption rates with ADP (State 3). (**A**) Mouse kidney mitochondria; (**B**) Rat brain mitochondria; (**C**) Rat heart mitochondria. The data are mean ± standard error. *** *p* < 0.01 Palm-Carnitine (P-C) vs. P-C + Succinate, P-C + Glutamate, and P-C + Pyruvate; * *p* < 0.05 Palm-Carnitine vs. P-C + Pyruvate (**A**) (*n* = 5). Redrawn from ref [75].

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
