# Peer review of "Metabolic Syndrome and β-Oxidation of Long-Chain Fatty Acids in the Brain, Heart, and Kidney Mitochondria"

_ijms, 2022, doi:10.3390/ijms23074047_

Round 1

Reviewer 1 Report

In the review article entitled: “Metabolic syndrome and β-oxidation of long-chain fatty acids in the brain, heart, and kidney mitochondria”, the authors provide a very comprehensive revision of the current knowledge on the implication of the β-oxidation of long-chain fatty acids in the pathobiology of the metabolic syndrome. The review article is well written and delivers an overarching message throughout the different sections of the article. The flow of the paper is consistent and the different sections are clear and balanced.

The authors provide the reader with the collected information about the topic but additionally debate in the literature. However, around 60% of the papers cited in this review were published more than 10 years ago, and only 9% of the papers cited were published in the last 3 years with only two of the cited papers being published in 2021. In this regard, authors should include more recent citations to highlight the timely relevance of the presented topic.

The figures of the paper are informative and clear. However, the quality should be improve since the letters in the graphs are a bit pixelated. Also in figure 5 the title of the y-axis of the graph C is a bit hidden by the graph in B. Similarly happens in figure 6, where the tittle of the y-axis of graph B is hidden by graph A.

Although grammatically well written I would suggest the authors to include the abbreviation MetS in the first line of the abstract after the words metabolic syndrome. The information on the meaning of the abbreviation T2D only appears in page 12 when the first time that this abbreviation is on page 4, please correct. For consistency, the title of the subsection 5.1 that is not written in bold as the others subtitles, could be corrected.

Author Response

Author's Reply to the Review Report.

Reviewer 1.

Comments and Suggestions for Authors

In the review article entitled: “Metabolic syndrome and β-oxidation of long-chain fatty acids in the brain, heart, and kidney mitochondria”, the authors provide a very comprehensive revision of the current knowledge on the implication of the β-oxidation of long-chain fatty acids in the pathobiology of the metabolic syndrome. The review article is well written and delivers an overarching message throughout the different sections of the article. The flow of the paper is consistent and the different sections are clear and balanced.

Point 1. The authors provide the reader with the collected information about the topic and debate in the literature. However, around 60% of the papers cited in this review were published more than 10 years ago, and only 9% of the papers cited were published in the last 3 years with only two of the cited papers being published in 2021. In this regard, authors should include more recent citations to highlight the timely relevance of the presented topic.

Answer to point 1. The reviewer asked a fascinating question, which cannot be answered quickly. The first author worked in Mitochondriology for a little more than 55 years and witnessed the Rise and the Fall of many hypotheses and changes in the methodology and instrumentation of mitochondrial research. First, science does not develop linearly, and the big scientific ideas tend to become wrong. We live right in a time of changing fundamental paradigms. There were times when the oxygraphs were common everywhere. Today, you will find hardly two-three papers where the respiratory activity of mitochondria has been studied correctly. The methodology used during the legendary 60s-80s turned unacceptable to study mitochondrial physiology and pathophysiology. Simple acceptance of the idea that mitochondria oxidize not one but a mixture of substrates and different organs and species utilize different substrate mixtures makes it very difficult to find support in the literature.

This is why we were able to find helpful information in the earlier publications. Currently, there are very few studies on mitochondrial respiration with the correct application of the new methodology. However, Reviewer 1 was correct in stressing the necessity to provide current publications related to the subject of β-oxidation of long-chain fatty acids. In the Conclusion, we included two references on the importance of β-oxidation of long-chain fatty acids for human development and functions.

Point 2. The figures of the paper are informative and clear. However, the quality should be improve since the letters in the graphs are a bit pixelated. Also in figure 5 the title of the y-axis of the graph C is a bit hidden by the graph in B. Similarly happens in figure 6, where the tittle of the y-axis of graph B is hidden by graph A.

Answer to point 2. We have checked all figures and redone those which required corrections and improvement, including Figures 5 and 6.

Point 3. Although grammatically well written I would suggest the authors to include the abbreviation MetS in the first line of the abstract after the words metabolic syndrome. The information on the meaning of the abbreviation T2D only appears in page 12 when the first time that this abbreviation is on page 4, please correct. For consistency, the title of the subsection 5.1 that is not written in bold as the others subtitles, could be corrected.

Answer to point 3. We thank the Reviewer 1 for the comments. We made all necessary corrections in accord with the Reviewer’s recommendations.

Reviewer 2.

Comments and Suggestions for Authors

I really congratulate the authors for their study.
I believe that the topic in question is urgent to address because the analysis of the publications related to the MetS still does not lead to a deep understanding of how and why the MetS develops. This Review aims to clarify the undisputed criteria for diagnosing the metabolic syndrome and also to sort out the controversies that demonstrate that the variation of metabolic characteristics in the MetS depend on environmental conditions and therefore differ in the various ethnic groups. This aspect, underestimated by many, gives the MetS all its complexity. The topic is well presented in this Review, in a comprehensive and above all linear way, interspersed with a detailed description of the role played by mitochondria. The reading is pleasant, albeit strictly scientific. 

Few changes / corrections to be made

Point 1. Regarding chapter 5.

  1. Properties of β-oxidation of long-chain fatty acids and generation of superoxide radicals by the kidney, brain, and heart mitochondria.
    Why does the next paragraph begin with 5.1, without a short title to accompany it?
    Next, the title of paragraph 5.2. is 'β-oxidation of long-chain fatty acid by isolated mitochondria of the kidney, brain and heart': what happened to the 'generation of superoxide radicals'? Has it disappeared?
    Regarding chapter 6, I would insert it as a paragraph of Chapter 5.

Answer to Point 2. Because all questions mentioned in Point 1 regarding Chapter 5 are interconnected, we answer them all. First, to avoid confusion, we reconsidered the numbering order and included former Section 6.1 to section 5 as 5.4, as Reviewer 2 recommended.

Point 2. Spelling errors: 
*5.2. β-Oxidation of long-chain fatty acid by the isolated kidney, brain, and heart mitochoindria. (line 386, in my version)
*7. The mechaisms of kidneys failure (line 467, in my version)
* dayly (line 470, in my version).

Answer to Point 2. All spelling errors have been corrected.

We that the Reviewers for their comments and recommendations.

Reviewer 2 Report

I really congratulate the authors for their study.
I believe that the topic in question is urgent to address because the analysis of the publications related to the MetS still does not lead to a deep understanding of how and why the MetS develops. This Review aims to clarify the undisputed criteria for diagnosing the metabolic syndrome and also to sort out the controversies that demonstrate that the variation of metabolic characteristics in the MetS depend on environmental conditions and therefore differ in the various ethnic groups. This aspect, underestimated by many, gives the MetS all its complexity. The topic is well presented in this Review, in a comprehensive and above all linear way, interspersed with a detailed description of the role played by mitochondria. The reading is pleasant, albeit strictly scientific. 
Few changes / corrections to be made
Regarding chapter 5,
5. Properties of β-oxidation of long-chain fatty acids and generation of superoxide radicals by the kidney, brain, and heart mitochondria.
Why does the next paragraph begin with 5.1, without a short title to accompany it?
Next, the title of paragraph 5.2. is 'β-oxidation of long-chain fatty acid by isolated mitochondria of the kidney, brain and heart': what happened to the 'generation of superoxide radicals'? Has it disappeared?
Regarding chapter 6, I would insert it as a paragraph of Chapter 5.
Spelling errors: 
*5.2. β-Oxidation of long-chain fatty acid by the isolated kidney, brain, and heart mitochoindria. (line 386, in my version)
*7. The mechaisms of kidneys failure (line 467, in my version)
* dayly (line 470, in my version)

Author Response

(The authors gave the same response as above.)
